# A randomized controlled study incorporating an electromechanical gait machine, the Hybrid Assistive Limb, in gait training of patients with severe limitations in walking in the subacute phase after stroke

Anneli Wall *, Jörgen Borg, Katarina Vreede, Susanne Palmcrantz

Division of Rehabilitation Medicine, Department of Clinical Sciences, Danderyd Hospital, Karolinska Institutet, Stockholm, Sweden

* anneli.wall@sll.se

## Abstract

Early onset, intensive and repetitive, gait training may improve outcome after stroke but for patients with severe limitations in walking, rehabilitation is a challenge. The Hybrid Assistive Limb (HAL) is a gait machine that captures voluntary actions and support gait motions. Previous studies of HAL indicate beneficial effects on walking, but these results need to be confirmed in blinded, randomized controlled studies. This study aimed to explore effects of incorporating gait training with HAL as part of an inpatient rehabilitation program after stroke. Thirty-two subacute stroke patients with severe limitations in walking were randomized to incorporated HAL training (4 days/week for 4 weeks) or conventional gait training only. Blinded assessments were carried out at baseline, after the intervention, and at 6 months post stroke. The primary outcome was walking independence according to the Functional Ambulation Categories. Secondary outcomes were the Fugl-Meyer Assessment, 2-Minute Walk Test, Berg Balance Scale, and the Barthel Index. No significant between-group differences were found regarding any primary or secondary outcomes. At 6 months, two thirds of all patients were independent in walking. Prediction of independent walking at 6 months was not influenced by treatment group, but by age (OR 0.848, CI 0.719–0.998, $p = 0.048$). This study found no difference between groups for any outcomes despite the extra resources required for the HAL training, but highlights the substantial improvements in walking seen when evidence-based rehabilitation is provided to patients, with severe limitations in walking in the subacute stage after stroke. In future studies potential subgroups of patients who will benefit the most from electromechanically-assisted gait training should be explored.

## Introduction

The most common acute manifestation of stroke is hemiparesis, which often has a strong negative impact on gait function [1, 2]. Although there is evidence that early onset, intensive,

**Data Availability Statement:** All relevant data are within the paper and its Supporting Information files.

**Funding:** This work was supported with grants from the Promobilia Foundation (16096, 17097, 17066) (AW), STROKE-Riksförbundet (na)(AW), NEURO Sweden (na) (AW), the Norrbacka-Eugenia Foundation (865/16) (AW), and a donation by Lars Hedlund (Karolinska Institutet Dnr 2-1582/2016) (JB). HAL suits were provided by Cyberdyne Inc., Japan. The study sponsors were not involved in the study design, data collection, analysis and interpretation of data, in writing the manuscript, or in the decision to submit the manuscript for publication. https://www.promobilia.se http://www.strokeforbundet.se https://neuro.se/ http://www.norrbacka-eugenia.se/

**Competing interests:** HAL suits were provided by Cyberdyne Inc. Japan and therefore Yoshiyuki Sankai and Hiroaki Kawamoto are listed as Investigators in the Study protocol (Supporting Information, S2). This does not alter our adherence to PLOS ONE policies on sharing data and materials.

**Abbreviations:** BBS, (Bergs Balance Scale); BI, (Barthel Index); BWS, (Body weight support); CGT, (Conventional gait training); CONV group, (Group receiving conventional gait training only); EAGT, (Electromechanically-assisted gait training); FAC, (Functional Ambulation Categories); FMA-LE, (Fugl-Meyer Assessment for control of voluntary movement in the lower extremity); HAL, (Hybrid Assistive Limb); HAL group, (Group receiving incorporated HAL training); ICF, (International Classification of Functioning, Disability and Health); NIHSS, (National Institute of Health Stroke Scale); T1, T2, T3, (Time point 1, 2, 3); 2MWT, (2-Minute Walk Test).

repetitive, and task-specific training of motor functions might accelerate recovery and improve the final outcome, including walking ability, after stroke [3–6], there is a need for further development of training methods in response to an increasing understanding of recovery and neuroplasticity [7, 8]. As recently highlighted, the dose (number of steps), intensity (heart rate and/or walking speed) and variability of task training also need careful consideration as they influence the result of training interventions [9]. Interventions applied to improve walking after stroke might include over-ground walking with assistance and/or ambulatory devices (such as walking aids or orthoses), strength and balance training, and the use of a treadmill with or without body weight support (BWS). Various technologies that are designed to enable higher dose and more intensive gait training programs have been explored, including electro-mechanically-assisted gait machines. Electromechanically-assisted gait training (EAGT) in combination with physiotherapy has been found to increase the odds of becoming independent in walking, and the greatest effect is seen when this is applied in the first three months after stroke onset in patients who were unable to walk [10]. Most previously evaluated gait machines have enabled gait training by use of automatic motion generated by the robots. However, active patient participation, real-time control strategies and voluntary muscle activation are considered important in order to achieve the best effects of EAGT [11, 12].

The HAL (Hybrid Assistive Limb) exoskeleton is a hybrid gait machine with a control system that aims to capture the wearer's voluntary actions (Fig 1A). HAL comprises two subsystems allowing both a voluntary and an automatic mode of action, namely Cybernic Voluntary Control (CVC) and Cybernic Autonomous Control (CAC). Both of these systems depend on the wearer's intention but in different ways. The control method and algorithm of the HAL system have been reported in detail previously [13–15]. In the CVC mode, HAL is triggered by the wearer's voluntary activation of their lower limb muscles, as recorded by surface electromyography (Fig 1B), to provide torque and support gait motions. In case of complete loss of voluntary activation of gait muscles, the CAC mode may be used. Predefined gait movements are then initiated and sustained based on the wearer's weight shifting as registered by force-pressure sensors in their shoes. The exoskeleton will swing the left leg when enough weight is put on the right leg in stance phase and vice versa. HAL is available in double and single-leg versions, and gait training with HAL may be performed on a treadmill or over ground with or without BWS.

The Hybrid Assistive Limb has been found feasible for gait training in patients with lower extremity paresis in the subacute stage after stroke, as well as in the long term phase [16]. Our piloting feasibility study [17], preceding the current study, found improvements in motor function, walking independence- and speed, balance, and activities of daily living performance after HAL training. The median (range) FAC-score improved from 0 (0–2) to 1.5 (1–4) from baseline to after the intervention. Later, additional studies have indicated beneficial effects on gait function and independence in walking [18–20] after HAL training, but the data do not allow any firm conclusions. Further, most previous studies of HAL were single group studies (without control treatment), why there was a need for blinded, randomized controlled studies.

Thus, the overall aim for the study was to evaluate the potential benefit on functioning and disability, of incorporating HAL training, compared to conventional gait training (CGT) only, as part of an inpatient rehabilitation program in the subacute stage after stroke.

## Aim and research questions

The aim was to explore potential differences at the end of the intervention and at long-term follow up when incorporating HAL training, as part of an inpatient rehabilitation program for patients with hemiparesis and severe limitations in walking after stroke.

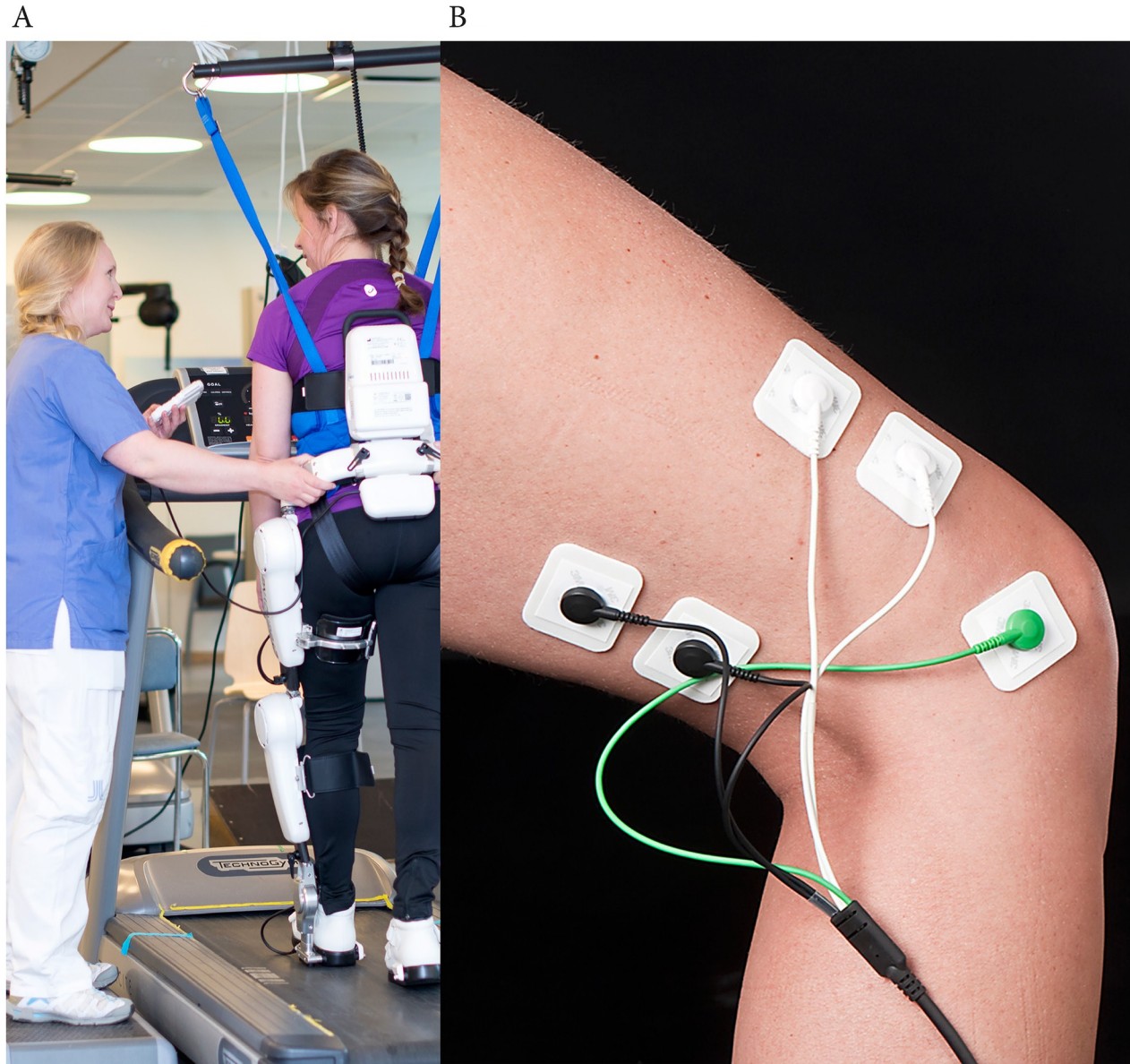

**Fig 1. The Hybrid Assistive Limb (HAL).** (A) Illustration of HAL training (B) surface electromyography to capture the wearer's voluntary muscle activation. Electrodes were affixed over the following muscles: biceps femoris (knee flexion), quadriceps vastus lateralis (knee extension) (in picture) and rectus femoris (hip flexion), gluteus maximus (hip extension) (not in picture). A reference electrode was placed over the lateral femur condyle. Photo: Johan Adelgren (A) and Carin Wesström (B). Consent for publication was obtained from the persons in the pictures.

The specific research questions were: Does 4 weeks of incorporated HAL training significantly improve 1) independence in walking, 2) movement functions, self-selected walking speed/endurance, balance, and/or self-care when compared to CGT only? We also wanted to explore which patient characteristics that might 3) predict independence in walking at 4 weeks (at the end of the intervention) and 4) at 6 months post stroke and 5) how patients rate their self-perceived effect of HAL training when compared to CGT.

## Methods

### Design

This prospective, randomized, open labeled, blinded evaluation (PROBE) study was conducted at the University Department of Rehabilitation Medicine at Danderyd Hospital, Stockholm, Sweden, admitting patients aged 18–67 years with moderate to severe acquired brain injury. The recruitment period lasted between February 2014 and December 2016, with the last follow-up performed in May 2017. All participants in this study expressed informed consent. The study was approved by the Regional Ethical Committee of Stockholm on November 13, 2013 (2013/1807-31/2) and was performed in compliance with GCP and the Declaration of Helsinki. The study was registered at ClinicalTrials.gov Identifier: NCT02410915, https://clinicaltrials.gov/ct2/show/NCT02410915. Registration was published in April 2015 after start of enrollment since registration was not part of the research routine at our department prior to this date. The authors confirm that all ongoing and related studies by our study group for this intervention are registered.

### Participants

Eligible patients were those who underwent team based, inpatient rehabilitation in the subacute stage after stroke. Inclusion criteria were ≤8 weeks since onset of ischemic or hemorrhagic stroke (verified by CT and/or MRI); inability to walk or in need of continuous manual support to walk due to lower extremity paresis (i.e. Functional Ambulation Categories (FAC) score 0–1) [21]; the ability to maintain a sitting posture with or without supervision for >5 minutes and, sufficient postural control to allow upright position in standing with aids and/or manual support; cognitive ability to understand training instructions as well as written and oral study information and express informed consent; and a body size compatible with the HAL suits. Exclusion criteria were cerebellar stroke, primary subarachnoid bleeding, contracture restricting gait movements at any lower limb joint, cardiovascular or other somatic condition incompatible with intensive gait training, and/or severe contagious infections.

### Randomization

A nurse, not otherwise involved in the study, manually randomized the patients according to a block design to either incorporated HAL training (HAL group) or CGT only (CONV group). For logistic reasons, randomization was performed in blocks of four (e.g. HAL, HAL, CONV, CONV or HAL, CONV, HAL, CONV etc.) to enable consecutive inclusion.

### Gait training with HAL

Training was performed using the single-leg version of HAL 4 days per week for 4 weeks (16 sessions in total). All sessions were conducted on a treadmill in combination with BWS and with one or two physiotherapists, educated in the HAL system, present (Fig 1A). Patients were encouraged to continue walking as far as possible, but at most for 60 minute's effective gait training time. Each session could at most proceed for 90 minutes, including time for putting on and taking off the suit, the gait training, and pauses at patients' request. The physiotherapists provided feedback through verbal instruction and/or by placing a mirror in front of the patients during the sessions. Patients used the handrail on the non-paretic side for support. During each session, output was adjusted via a control unit by which the therapist adapted the level of assistance over the hip and knee joint. The settings were individually adapted as the training progressed and optimized based on continuous observational gait analysis [22] in order to achieve a gait pattern as close to normal gait as possible (i.e. avoiding compensatory

movements such as circumduction). During the first session, BWS was set to 30% of the participant's weight, and training was performed using the CAC mode at both the hip and knee joint. The CVC mode was then planned to be used during the following 15 sessions. The initial speed of the treadmill was individually adjusted but started at lowest with a speed of 0.5 km/h. As the participants improved in walking ability, the amount of BWS and assistance was reduced, and the treadmill speed increased based on the physiotherapist's continuous observations. As in conventional gait training the HAL training was set to be challenging and not more assistance than needed was provided. All HAL sessions were documented using a standardized protocol, including the individual settings and training performance. After finishing the intervention period, the conventional team-based, individually adapted training program continued until discharge.

## Conventional training

Conventional team-based training was evidence based, individualized, and performed according to current best practice for inpatient rehabilitation after stroke [4] on weekdays, 5 days/week. The conventional team-based training was offered to both study groups, included physiotherapy training, most often daily for 30–60 minutes, and comprised e.g. training of motor function in the upper and lower extremity, trunk control, transferring oneself, and gait. The CGT could include standing, weight shifting, stepping, over-ground walking with manual assistance and/or assistive devices (such as walking aids and braces) as well as the use of a treadmill with/without BWS. Study group allocation was not planned to affect other team-based interventions. Physiotherapy sessions including CGT were documented by the patient's team physiotherapist in the patient's medical record regarding type of gait training and estimated time and distance walked.

## Clinical assessments

A blinded physiotherapist, experienced in stroke rehabilitation, performed clinical assessments at three time points–before (T1) and after (T2) the intervention and at 6 months after stroke onset (T3). The applied assessment instruments are valid and reliable and cover several aspects of body function and activity according to the ICF (International Classification of Functioning, Disability and Health) [23]. The primary outcome was the Functional Ambulation Categories (FAC) score [21]. FAC allows scoring of walking ability on a six-grade scale ranging from independent walking outdoors, on stairs, and on unlevel surfaces (score 5) to non-functional walking (score 0). A score of 4 corresponds to independent walking on level surfaces. A score of 1–3 corresponds to dependent walking in need of varying degree of manual support (score 1–2) or supervision (score 3). Thus, the FAC can be dichotomized into dependent in walking (FAC <4) and independent in walking (FAC ≥4) [24]. Other assessments performed by the blinded physiotherapist were the Fugl-Meyer Assessment for control of voluntary movement in the lower extremity (FMA-LE Motor) (motor function domain, 0–34 points) [25], the 2-Minute Walk Test at self-preferred speed (2MWT) [26], the Bergs Balance Scale (BBS) (0–56 points) [27, 28], and the Barthel Index for dependence in activities of daily living (BI) (0–100 points) [29]. A blinded senior consultant assessed stroke severity at T1 using the NIH Stroke Scale (NIHSS) for stroke severity (0–42points) [30]. After the intervention period, the same senior consultant (no longer blinded) administered a study-specific questionnaire assessing patients' self-perceived beneficial effect of their gait training. The questionnaire comprised a Likert scale of 0–10 (where 0 equals *none at all* and 10 equals *largest possible*. All data were documented by use of standard forms (paper and pencil) and kept in individual Case Report Forms. In addition, FAC was assessed weekly by the patient's physiotherapist at the ward and documented in the patient's medical record.

## Power calculation and sample size

The power calculation was based on available data from previous studies in this area [17, 31] using the FAC score. The score has been found to have good responsiveness and concurrent validity of each step along the scale in relation to functional mobility, walking distance and walking speed [32]. With FAC as the primary outcome measure and an expected difference of 1 level between groups, SD 1, an alpha value of 0.05, and a power of 80%, 16 patients were required in both groups. We assessed the risk of loss to not exceed two patients per treatment group and therefore intended to include a total of 36 patients, 18 in both groups.

## Statistics

Descriptive statistics and tests for normality were examined using skewness, boxplots, and Q-Q-plots. Data management and analysis were performed in SPSS (IBM SPSS Statistics 25) with significant levels set to $p < 0.05$ (two-tailed). Data were analyzed as an all-available-data analysis.

Between-group differences were tested with the Mann-Whitney U-test for age, stroke severity, mean distance walked during CGT, number of CGT sessions, FAC, BBS, FMA-LE Motor, 2MWT, BI, and self-reported perception of gait training; with the Independent sample t-test for days from stroke, total number of gait training sessions, and percent of sessions with BWS; and with the Chi-square test/Fisher's exact test for gender, diagnosis, paretic side, and proportion of patients with independent walking and walking speed classification belonging. Walking speed (2MWT) was classified according to Perry et al. [33] as household ambulation ($<$0.4 m/ s), limited community ambulation (0.4–0.8 m/s), or community ambulation ($>$0.8 m/s). The Wilcoxon Signed-Rank Test was used for comparison of distance walked during training within the HAL group.

For prediction of independence in walking at T2, we performed ordinal regression. At T3 we performed binary logistic regression with FAC as the dependent variable because the data could now be dichotomized as dependent (FAC $<$4) or independent (FAC $\geq$4) and with the variables group, sex, age, diagnosis, paretic side, FMA-LE Motor at T1, BBS at T1, and NIHSS at T1 as the independent variables. A univariate approach was used, and significant ($p < 0.05$) predictors were combined in a multivariable model (using forward conditional).

# Results

## Patient characteristics

During the recruitment period, 273 patients (69% men and a mean age of 51 years (SD 11)) were screened for eligibility (Fig 2). Most common reasons for exclusion were diagnoses other than stroke and/or a FAC score $>$1. Sixteen patients in both groups completed the intervention period. Two patients were lost to follow up (at T3), both in the CONV group (due to private or to medical factors). The NIHSS score at baseline corresponded to moderately severe stroke in both groups [34, 35]. There was no statistically significant difference ($p > 0.05$) in any patient characteristics at baseline. Patient characteristics are shown in Table 1.

## Gait training with HAL

HAL training was conducted by two physiotherapists present during all sessions, except for a few exceptions where only one therapist was needed. Thirteen patients performed all 16 HAL sessions, two performed 15 sessions, and one performed 14 sessions. Reasons for cancelling sessions were logistics (n = 2) or medical conditions (n = 2). Apart from the first session, four patients had to use the CAC mode in 1–3 additional sessions or part of a session at the hip

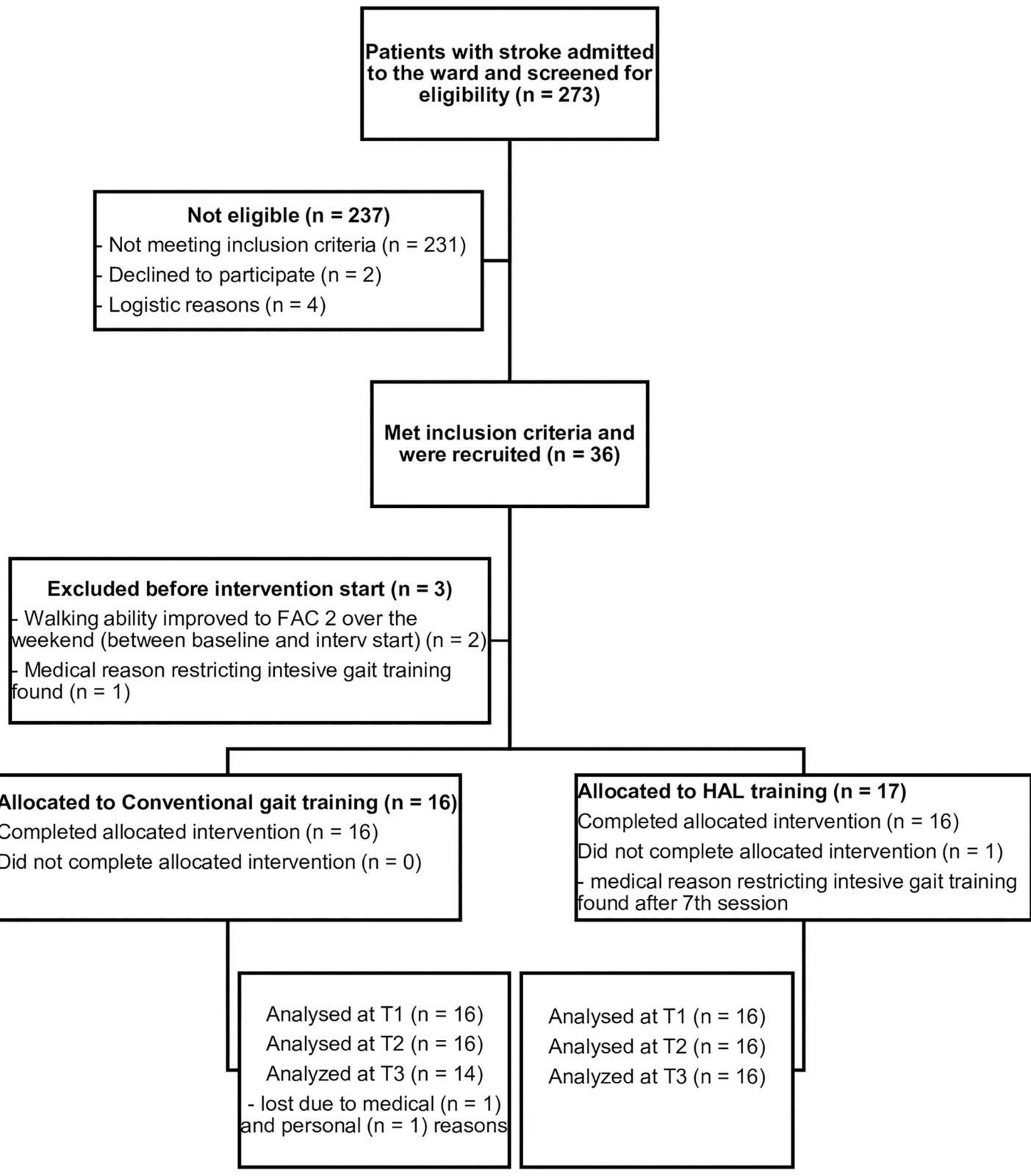

**Fig 2. CONSORT flowchart of recruitment of patients.** T1: time point 1 (Baseline); T2: time point 2 (after intervention); T3: time point 3 (6 months post stroke); FAC: Functional Ambulation Categories.

**Table 1. Patient characteristics.**

|  | HAL (n = 16) | CONV (n = 16) |
|---|---|---|
| *Age (years)* |  |  |
| *Median [IQR]* | 55 [48.25;62.5] | 57.5 [54.25;60.75] |
| *Sex* |  |  |
| *Male n (%)* | 13 (81) | 13 (81) |
| *Days from stroke* |  |  |
| *Mean (SD)* | 32 (15) | 36 (16) |
| *Stroke characteristics* |  |  |
| *Stroke Type* |  |  |
| *Hemorrhage n (%)* | 5 (31) | 8 (50) |
| *Infarction n (%)* | 11 (69) | 8 (50) |
| *Paretic side* |  |  |
| *Left n (%)* | 13 (81) | 8 (50) |
| *Stroke severity (NIHSS)* |  |  |
| *Median [IQR]* | 11.5 [8.25;14.5] | 13 [10;18]† |
| *Baseline FAC* |  |  |
| *FAC 0 n (%)* | 11 (69) | 11 (69) |

HAL: HAL group, CONV: Conventional group, NIHSS: NIH Stroke Scale; FAC: Functional Ambulation Categories; SD: Standard deviation; IQR: Inter-quartile range.

† n = 15

and/or knee joint due to insufficient surface electromyography signals or disconnected electrodes. One patient used the CAC mode (mostly at the hip joint) during 11 sessions due to technical issues with the suit (related to the use of the CVC mode). No adverse events occurred. The settings were individually adjusted, and assistance was decreased continuously, as participants improved. At the last session the amount of BWS had decreased from the initial 30% of the patient's body weight to 19% in mean (SD 5). At the initial HAL sessions, the mean walking speed was 0.8 km/h (session 1) and 0.9 km/h (session 2), and during the last sessions (14–16) it was 1.5 km/h. The mean distance at the initial session (performed with CAC mode) was 249 m (SD 148), and at the initial session with CVC mode (i.e. session 2) it was 236 m (SD 91). The mean distances walked during the last two sessions were 845 m (SD 429) and 933 m (SD 476), respectively. The overall mean distance walked per HAL session was 619 meters (SD 368). Data on distance walked for all sessions are presented in Fig 3.

## Conventional gait training

The estimated distance walked during CGT was recorded in the medical records in 80% of all sessions (CONV group 76%, HAL group 84%). Data were missing at random during the intervention period. There was no significant difference in distance walked during the CGT sessions between the two groups ($p = 0.078$) (CONV group 60 m (IQR 22;138); HAL group 30 m (IQR 15;50)). However, in the HAL group, there was a significant difference in walking distance per session during the CGT compared to during the HAL training ($p = 0.001$), with longer distances during the latter. We found a significant difference in the number of CGT sessions performed (CONV group 10.5 (IQR 8;14.5); HAL group 6 (IQR 5;8.75) $p = 0.003$), with approximately one more CGT session per week for the CONV group. The total number of gait training sessions, including HAL training, was, however, greater in the HAL group, who performed a median of 11 sessions more compared to the CONV group ($p < 0.001$)

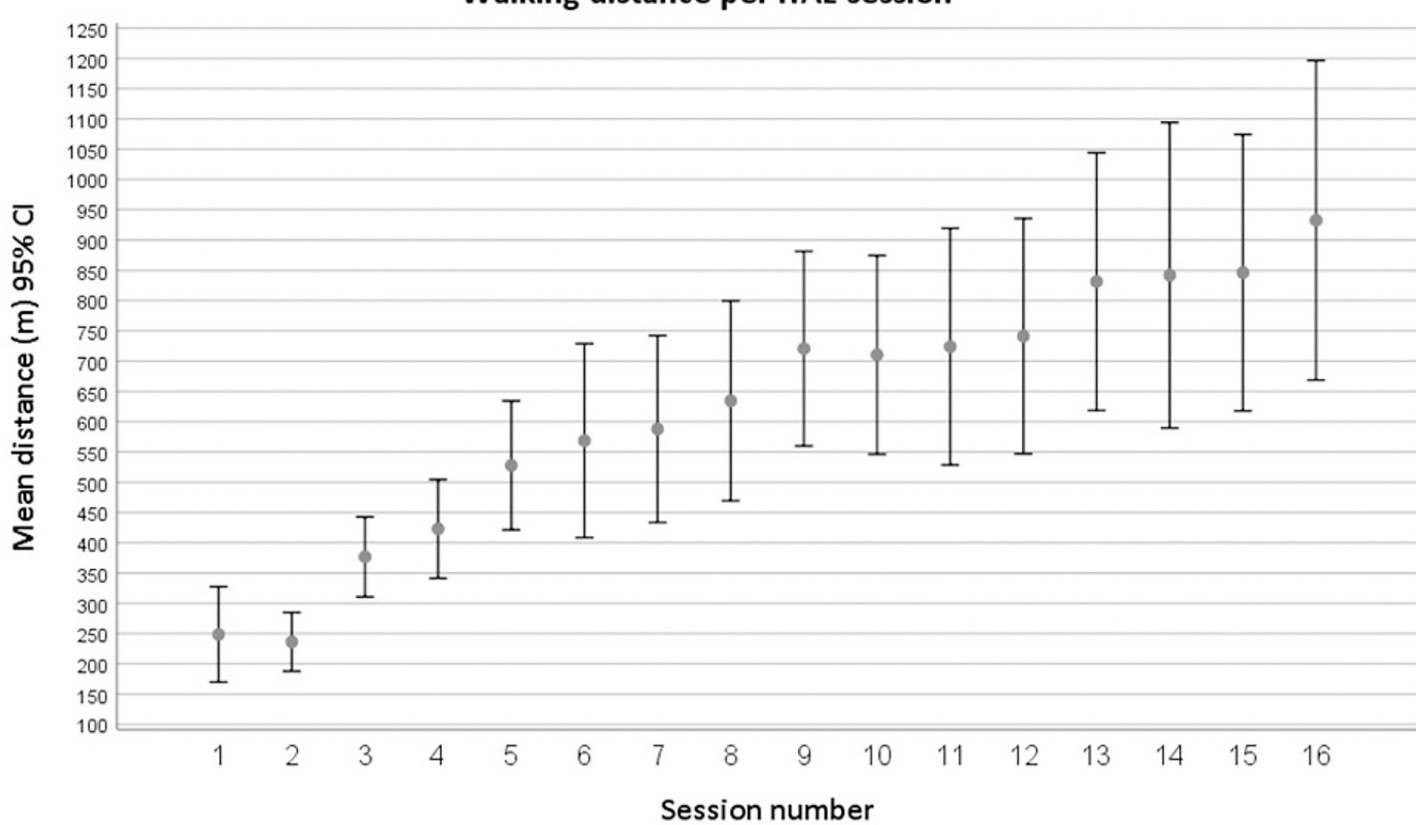

**Fig 3. Distance (meters) walked during HAL sessions (1–16).** Distances displayed as means and 95% confidence intervals (CI). At session 6, 13, 14 and 16, 15 patients participated. At all other sessions 16 patients participated.

(CONV group 10.5 (IQR 8;14.5), HAL group 22 (IQR 19.5;24)). The CONV group performed 29% (IQR 17;41) of the individual CGT sessions using BWS and treadmill compared to 73% (IQR 66;82) in the HAL group (both CGT and HAL training) ($p < 0.001$).

### Independence in walking

There were no significant differences in change between the three time points in independence in walking (FAC score) between the groups ($p > 0.05$) according to the Mann-Whitney U-test (Table 2). Data on patients classified as dependent (FAC $< 4$) or independent in walking (FAC $\geq 4$) at T1, T2, and T3 are presented in Fig 4. There was no significant between-group difference in the proportion of patients with independent walking at T2 ($p = 1.000$) or T3 ($p = 1.000$). Patients improved continuously during the intervention period, and the improvements were similar in both groups with no significant difference in the weekly FAC score between groups (Week 2 $p = 0.752$, Week 3 $p = 0.800$, Week 4 $p = 1.000$).

### Secondary outcomes

There were no significant differences in change between the three time points in any of the secondary clinical outcomes between the groups ($p > 0.05$) (Table 2). In addition, there was no significant difference in walking speed classification (according to Perry et al. 1995) between the groups after the intervention (T2) ($p = 0.525$) or at follow up (T3) ($p = 0.703$). In the HAL

**Table 2. Outcome measures.**

| Outcome measure (subjects per group HAL/CONV) | HAL | CONV | Between groups (p-value) |
|---|---|---|---|
| Independence in walking (FAC) | | | |
| FAC T1 (16/16) | 0 [0;1] | 0 [0;1] | 1.000 |
| FAC T2 (16/16) | 2 [1.25;3] | 2.5 [1.25;3] | |
| FAC T3 (16/14) | 4 [3;5] | 4 [2.75;5] | |
| ΔT1-T2 (16/16) | 2 [1;2.75] | 2 [1;3] | 0.926 |
| ΔT2-T3 (16/14) | 2 [1;2] | 1.5 [0.75;2.25] | 0.473 |
| ΔT1-T3 (16/14) | 4 [3;4.75] | 4 [2.75;4] | 0.728 |
| Balance (BBS) | | | |
| BBS T1 (16/16) | 8 [5.25;12.75] | 8 [5.25;14.75] | 0.696 |
| BBS T2 (16/16) | 22 [10.25;40.25] | 22.5 [11.5;35] | |
| BBS T3 (16/14) | 45 [29.75;52] | 45.5 [25.75;53.25] | |
| ΔT1-T2 (16/16) | 8.5 [6;31.75] | 14 [4.25;20] | 0.780 |
| ΔT2-T3 (16/14) | 15.5 [3;27.25] | 18 [6.75;25.5] | 0.886 |
| ΔT1-T3 (16/14) | 34.5 [17.75;41,5] | 32.5 [19.75;41.25] | 0.918 |
| Motor function (FMA-motor) | | | |
| FMA T1 (16/16) | 7.5 [4;13.5] | 7.5 [4;17.75] | 0.590 |
| FMA T2 (16/16) | 14 [5.25;21.5] | 14.5 [4.25;27] | |
| FMA T3 (16/14) | 21 [13.25;25.25] | 15.5 [11;29.5] | |
| ΔT1-T2 (16/16) | 3 [2;9.75] | 1 [-0.75;6.75] | 0.224 |
| ΔT2-T3 (16/14) | 4 [2;7.5] | 2 [0.75;9.25] | 0.355 |
| ΔT1-T3 (16/14) | 12 [5.5;15] | 7 [3;17.25] | 0.580 |
| Activities of Daily Living (BI) | | | |
| BI T1 (16/16) | 35 [30;50] | 40 [35;45] | 0.590 |
| BI T2 (16/16) | 60 [42.5;65] | 57.5 [45;65] | |
| BI T3 (16/14) | 90 [71.25;98.75] | 90 [62.5;96.25] | |
| ΔT1-T2 (16/16) | 20 [15;28.75] | 12.5 [10;25] | 0.149 |
| ΔT2-T3 (16/14) | 27.5 [11.25;38.75] | 30 [17.5;35] | 0.854 |
| ΔT1-T3 (16/14) | 45 [40;58.75] | 47.5 [28.75;55] | 0.525 |
| Walking speed/endurance (2MWT) | | | |
| 2MWT T1 (16/16) | 4 [0;7] | 2 [0;13.625] | 0.926 |
| 2MWT T2 (16/16) | 20.75 [8.5;44.75] | 20.75 [7.25;60.25] | |
| 2MWT T3 (16/14) | 39.5 [26.5;108.125] | 69.5 [21.875;137.125] | |
| ΔT1-T2 (16/16) | 14.75 [8.375;38] | 12.5 [7.125;51] | 0.838 |
| ΔT2-T3 (16/14) | 22.750 [8.25;50.875] | 37 [11.375;79.625] | 0.728 |
| ΔT1-T3 (16/14) | 35.75 [26.5;101.125] | 68.5 [21.5;121.625] | 0.728 |

Values presented as median and inter-quartile range [IQR]. HAL: HAL training group; CONV: Conventional gait training group; FAC: Functional Ambulation Categories; BBS: Berg Balance Scale; FMA: Fugl-Meyer Assessment; BI: Barthel Index; 2MWT: 2 Minute Walk Test; T1: time point 1 (Baseline); T2: time point 2 (after intervention); T3: time point 3 (6 months post stroke); Δ: change. P-values according to the Mann-Whitney U-test.

group 12 patients (75%) were classified as household ambulators and 4 (25%) as limited community ambulators at T2. In the CONV group the corresponding numbers were 11 (69%) and 3 (19%) respectively and with 2 patients (13%) classified as community ambulators. At T3 9 (56%) were household ambulators, 2 (13%) were limited community ambulators and 5 (31%) were classified as community ambulators, in the HAL group. In the CONV group the corresponding numbers were 6 (43%), 3 (21%) and 5 (36%).

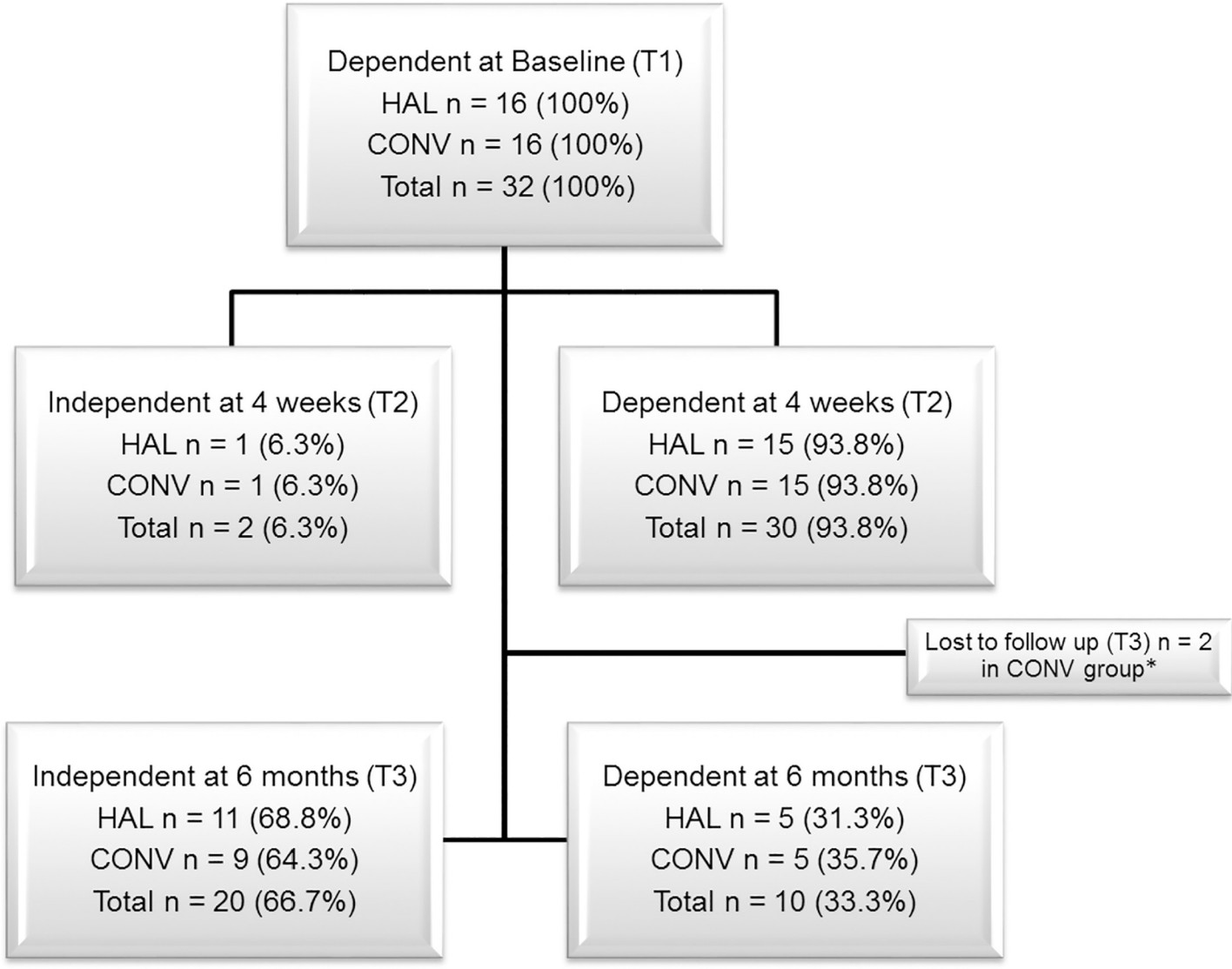

**Fig 4. Patients with dependent or independent walking.** Patients with dependent (0–3) or independent walking (4–5) according to Functional Ambulation Categories at baseline (T1), after intervention (T2) and at 6 months post stroke (T3). HAL: HAL group; CONV: Conventional group; T1: time point 1 (Baseline); T2 time point 2 (after intervention); T3: time point 3 (6 months post stroke). *Lost to follow up (T3) scored FAC 2 and 3 at T2.

### Prediction of independence in walking

Ordinal regression showed that the odds of having a higher FAC score after the intervention (at T2) was not influenced by intervention group (OR 1.095, $p$ = 0.888). At 6 months post stroke (T3), the odds of being independent in walking (FAC ≥4) were not influenced by intervention group, sex, diagnosis, paretic side, initial motor function (FMA motor at T1), or initial balance (BBS at T1) (Table 3). However older age and more severe stroke (according to NIHSS at T1) decreased the odds of being independent in walking at T3. Age and NIHSS were thus included in a model together with Age*NIHSS, but all were found to be non-significant. By using the forward conditional approach in the multivariate model, we found that only Age was left in the model suggesting that Age is a key predictor (OR 0.848, CI 0.719–0.998, $p$ = 0.048). All patients younger than 54 years of age were independent in walking at 6 months post stroke.

**Table 3. Odds of being independent in walking (FAC ≥4) at 6 months post stroke using univariate binary logistic regression.**

| Independent variables (univariate) | OR | 95% CI | p-value |
|---|---|---|---|
| Group | 0.818 | 0.179–3.744 | 0.796 |
| Sex | 1.000 | 0.150–6.671 | 1.000 |
| Diagnosis | 1.909 | 0.380–9.590 | 0.432 |
| Paretic side | 4.000 | 0.765–20.92 | 0.101 |
| Motor function (FMA motor T1) | 1.099 | 0.942–1.282 | 0.229 |
| Initial balance (BBS T1) | 1.107 | 0.940–1.304 | 0.225 |
| Age | 0.848 | 0.719–0.998 | 0.048* |
| Stroke severity (NIHSS T1) | 0.793 | 0.635–0.989 | 0.040* |

OR: Odds ratio; CI; Confidence Interval.

* Significant and thus included in multivariate analysis (forward conditional).

## Self-perceived effect of gait training

There was no significant difference between groups for self-perceived beneficial effect of the CGT practice ($p = 0.292$). Patients in the HAL group reported significantly more beneficial effect of HAL training compared to the CGT ($p = 0.031$) (Table 4).

## Discussion

The main finding of this PROBE study was that there was no significant difference after 4 weeks of HAL training with regard to independence in walking, movement function, self-selected walking speed, balance, or self-care when added to conventional training in the sub-acute phase for patients with severe limitations in walking 5 weeks after stroke. The odds of being independent in walking at 6 months post stroke were influenced by age, but not by treatment. These results indicate that incorporating HAL training does not influence the beneficial outcome seen in these younger patients who received evidence-based, CGT in a specialized neurorehabilitation setting due to severely limited walking ability after stroke.

In stroke rehabilitation, even though issues remain, there appears to be a dose-response relationship [36–38] where increased practice of walking and activities related to walking in the post-acute phase after stroke results in better gait outcomes such as walking ability and speed [39]. Our hypothesis was that HAL training in combination with CGT would improve walking ability more than CGT alone by allowing both larger dose, higher training intensity, voluntary-driven guided force, and sensory feedback. Although our hypothesis was not confirmed, our results indicate that HAL training might provide a way of enabling longer walking distances, more steps, and higher intensity. Yet, the extra time for rehabilitation interventions needed in order to optimize the beneficial effect on activity performance in upper and/or

**Table 4. Self-perceived beneficial effect of the gait training.**

| Questions | HAL (n = 13) | CONV (n = 9) |
|---|---|---|
| To what extent have you experienced a beneficial effect from the gait training? | 8.3 [6.7;9.5] | 9.5 [8;10] |
| To what extent have you experienced a beneficial effect from the gait training with HAL? | 9 [7.9;10] | n.a. |

Zero equals *none at all* and ten equals *largest possible*. Values presented as median and inter-quartile range [IQR].
HAL: HAL group, CONV: Conventional group.

lower limb is proposed to be as large as 240% [37]. Another possible explanation for the lack of group differences in the present study might be that the design of HAL training on a treadmill and with BWS provides less variation of the task and requires less balance control compared to progressive over-ground walking. To promote adequate motor learning, variation of the task is important and should be incorporated in the training [38, 40]. Skills obtained in one setting (such as on a treadmill) might not be automatically transferred to another setting (such as over-ground walking). In our study the patients who performed HAL training spent proportionally more of their gait training time on a treadmill than the CONV group (73% and 29%, respectively). Future studies should consider EAGT in combination with limited/well balanced use of BWS and systems that enable an early start to over-ground gait practice and task-specific training of activities of daily living in non-ambulatory individuals (post stroke).

In the present study, most patients had no functional walking ability at baseline, and the Fugl-Meyer motor domain and NIHSS confirmed that patients had remaining moderate to severe lower extremity motor impairment. Nevertheless, at 6 months post stroke a total of 67% were independent in walking (FAC ≥4) in our study, which is in accordance with [41] and even superior to [2] findings in earlier studies. For example, Kollen et al. [42] reported that in mean 50% of the possible improvement (i.e. maximal score minus the score at inclusion) on the FAC score was achieved at 8–9 weeks after stroke onset (corresponding to our post intervention measurement, T2), and 76% at 6 months after stroke (corresponding to our T3). The corresponding figures in our study were 47% (T2) and 82% (T3). Thus, although patients in our study represent a more severe subgroup with limited mobility and were included later after stroke onset than in previous studies [2, 41, 42], the degree of improvement was similar. To what extent the improvements reflect true recovery or compensation remains to be distinguished [43] and might be elucidated by available gait analyses data from a subgroup of the study participants. Nevertheless, these results speak to the need and clear benefit of providing evidence-based specialized rehabilitation interventions for this sub-group of patients who are at risk of facing long-term disability.

As pointed out, our study group was relatively young compared to the overall stroke population, but still we identified an even younger subgroup, below 54 years of age, who all were independent in walking at 6 months post stroke. Age has previously been found to be important for improvement in walking independence [41, 42] and has together with stroke severity (NIHSS score) been found to be useful in predicting recovery of upper limb function after stroke (Predict Recovery Potential (PREP2) algorithm) [44]. It is plausible that potential differences between groups in our study were masked by this factor, and thus future studies might benefit from including a larger study sample that allows sub-group analysis of patients in different age groups. Other aspects that have been found to be important when predicting independence and time to independence in walking are trunk control and lower limb muscle strength, especially hip extension which have been suggested as key factors regardless of age [24, 45]. Patients in the subacute stage after stroke, who have good residual trunk control have also been found to be good responders to EAGT using the end-effector-based Gait Trainer [46]. All patients in our study had sufficient trunk control to maintain balance in sitting, but hip extension muscle strength was not measured. Thus, future intervention studies should consider these variables for stratification and/or subgroup analyses.

This is to the best of our knowledge the first assessor-blinded randomized controlled study using HAL for gait training after stroke with a 6-months follow-up. Previous randomized clinical studies in the subacute-phase after stroke both agree with and contradict the results in the present study. Significant improvements in independent walking (FAC) after HAL training compared to CGT after both a 4-week intervention [20] and at 2-months follow-up [18] have been seen among patients with a higher initial FAC score than in the present study (FAC 2).

However, secondary outcome measures like walking speed did not differ between groups [18]. Another non-randomized clinical study with a 5 week intervention, in patients with an initial FAC score of 3, performed around 5 months post stroke, found no significant difference in walking independence but greater improvements in maximum and self-selected walking speed in the HAL group [19]. In these studies, the groups were small (n = 6–12) and the assessors were not blinded. A study with a larger sample (n = 53) [47] suggested that improvements in walking speed after HAL training in the acute phase after stroke are less in patients with severe hemiparesis compared to those with less severe hemiparesis, but that study did not compare HAL training to conventional rehabilitation. Further studies with sufficient statistical power are needed to address the question about who will benefit from HAL training based on their initial level of dependence in walking.

A recent review [10] including a number of different electromechanically-assisted training devices for walking after stroke, found moderate-quality evidence that non-ambulatory patients benefit from EAGT combined with conventional physiotherapy compared to conventional physiotherapy only. However, no clear conclusion regarding a long-lasting effect of the use of electromechanical devices is provided, which is in line with the results of the present study. A closer comparison of studies included in the review [48–54], and additional studies [55, 56] that are comparable to our study design and setting, exhibit both concordant and conflicting results.

A number of studies [48–50, 54–56] have found results corroborating ours with no significant difference in independence in walking [48–50, 54, 55], walking speed [54, 56], gait performance [55], balance or movement function [49] after EAGT (with the Lokomat) compared to CGT in patients with dependent walking at baseline [49, 50, 54–56]. However, one study reported higher walking speed and longer distances achieved during EAGT [54] and two studies [48, 49] reported improvements in cardiorespiratory fitness compared to the conventional group.

Other studies present results conflicting with the results from our study. Studies in non-ambulatory patients in the subacute stage after stroke [51–53] have found a greater improvement in FAC after robotic gait training compared to conventional training. However, in the study by Kim et al. [51] none of the groups reached independent walking (i.e. FAC ≥ 4) at 6 months follow up. In addition, the post-intervention FAC scores in Ochi et al. [52] and Swartz et al. [53] are similar to the FAC scores in our HAL group, but lower for their conventional groups compared to our conventional group. Furthermore, there was no significant difference in other gait outcomes (2MWT, 10MWT) in those studies [52, 53]. A possible explanation for the diversity in results in previous EAGT studies might be small study samples and low study power, heterogeneity in the stroke population, lack of blinding, differences in the amount and intensity of training administered, and the content of the conventional/control therapy.

Most studies discussed above had dose-matched interventions (i.e. the two groups received the same amount of training time). In our study, HAL training was incorporated in the conventional team-based training. However, as presented above, the number of CGT sessions were significantly fewer and although non-significant the distances walked during the CGT sessions shorter in the HAL group. Possible reasons for this might have been fatigue, limitations of available training time due to other team-based interventions (such as speech- and/or occupational therapy) or that some of the CGT were replaced with HAL training. As pointed out by others, robotic systems should support the rehabilitation program and not replace the therapists and/or the conventional therapy [57], but the optimal mix, duration, and number of training sessions remain to be explored.

Another issue relates to the cost-effectiveness of EAGT. It has been suggested that the use of EAGT might reduce therapist burden, but the cost-effectiveness is yet to be investigated [10].

However, in the present study two therapists were needed during most of the HAL training sessions due to severe limitation in walking among the patients.

Even though we found no additional effect of HAL training on the primary or secondary outcomes, our study indicates other areas that may deserve attention in future studies. Patients walked more than 500 m longer distances during HAL training than during CGT, why further studies may consider potentially beneficial effects on cardiovascular and respiratory functions of training with the HAL system. We also recognize a need for individually designed interventions that consider the large variation between individuals with regard to both the expected recovery and response patterns to specific interventions. Thus, a larger sample size would produce more sufficient power and the possibility of subgroup analysis, preferably based on age, initial motor function, and level of independence in walking. We also suggest that dose-matched interventions with gait training performed mainly over ground should be considered in future controlled studies in this area.

## Limitations

There are certain limitations in this study. Patients were recruited from the Department of Rehabilitation Medicine at Danderyd Hospital, where the majority of patients referred are aged 18–67 years. Stroke is more common in the older population, and we can therefore not generalize our findings to the whole stroke population. Women usually suffer stroke at older age than men, which might have caused an overrepresentation of men in our study; nevertheless, the genders were evenly distributed between the two intervention groups. Because the sample size was small, caution must be applied when interpreting our results. Power calculation was performed for our primary outcome; however, two patients were lost to the 6 months follow up, which reduced the power. In addition, between-group differences in the primary outcome were less and the variance was greater than expected. Power calculation for secondary outcomes was not performed, but our results might be useful for future studies in this respect. One other issue that needs consideration is the difference in the amount of CGT carried out by the groups, with fewer sessions in the HAL group. This might have had an impact on the ability to transfer skills obtained in HAL training to walking over ground. Future studies should consider the best possible matching of gait training time offered to patients. On the other hand, patients in our study were almost entirely unable to walk at baseline, and we found HAL useful to start intensive training of gait early, as also demonstrated by the longer distance walked during each session compared to the CGT. The FAC score is a commonly used outcome after gait interventions. The ceiling effect of this measurement might be a problem, but this was considered small in this study where only 12 patients (40%) received a score of 5. Finally, between T2 and T3 all patients were discharged from the inpatient ward and were thereby not monitored regarding rehabilitation training. Consequently, it was not possible to control for potential differences in outpatient rehabilitation during this period and its impact on the results at the T3 assessments. Although the health care system in Stockholm, Sweden, offers quite similar outpatient rehabilitation after stroke, the rehabilitation might have differed in intensity and content.

## Conclusion

This study found no difference between groups for any outcomes after the intervention or at 6 months post stroke, despite the extra resources required for the HAL training. In these younger patients with hemiparesis and severe limitations in walking in the subacute stage after stroke independence in walking at 6 months was not related to group allocation but to a younger age. The results should be interpreted with caution considering that the sample size was

small, and that between-group differences were less and variance greater than expected. The findings suggest that further explorations of who in the heterogeneous stroke population will benefit the most from EAGT should be addressed in future studies.

## Supporting information

**S1 Appendix. CONSORT checklist.**
(PDF)

**S2 Appendix. Study protocol.**
(PDF)

**S3 Appendix. Dataset.**
(XLSX)

## Acknowledgments

The authors would like to acknowledge associate professor Disa Sommerfeld for her contribution to this work.

## Author Contributions

**Conceptualization:** Anneli Wall, Jörgen Borg, Susanne Palmcrantz.

**Formal analysis:** Anneli Wall, Susanne Palmcrantz.

**Funding acquisition:** Anneli Wall, Jörgen Borg, Katarina Vreede, Susanne Palmcrantz.

**Investigation:** Anneli Wall, Katarina Vreede.

**Methodology:** Anneli Wall, Jörgen Borg, Susanne Palmcrantz.

**Project administration:** Anneli Wall.

**Resources:** Anneli Wall.

**Supervision:** Jörgen Borg, Susanne Palmcrantz.

**Validation:** Anneli Wall, Jörgen Borg, Katarina Vreede, Susanne Palmcrantz.

**Visualization:** Anneli Wall.

**Writing – original draft:** Anneli Wall.

**Writing – review & editing:** Jörgen Borg, Katarina Vreede, Susanne Palmcrantz.

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
