## [Decision Letter · Decision Letter 0]

1 Oct 2019

PONE-D-19-15743

A randomized controlled trial incorporating an electromechanical gait machine, the Hybrid Assistive Limb, in gait training of patients with severe limitations in walking in the sub-acute phase after stroke

PLOS ONE

Dear Ms. Wall,

Thank you for submitting your manuscript to PLOS ONE. After careful consideration, we feel that it has merit but does not fully meet PLOS ONE’s publication criteria as it currently stands. Therefore, we invite you to submit a revised version of the manuscript that addresses the points raised during the review process.

Please address the concerns from the reviewers particular the sample size and adjustment for multiple testings. The conclusion and the results should parallel each other

We would appreciate receiving your revised manuscript by Nov 15 2019 11:59PM. To enhance the reproducibility of your results, we recommend that if applicable you deposit your laboratory protocols in protocols.io, where a protocol can be assigned its own identifier (DOI) such that it can be cited independently in the future. For instructions see: http://journals.plos.org/plosone/s/submission-guidelines#loc-laboratory-protocols

We look forward to receiving your revised manuscript.

Kind regards,

Thanh G Phan, PhD

Academic Editor

PLOS ONE

**Journal Requirements:**

2. Thank you for submitting your clinical trial to PLOS ONE and for providing the name of the registry and the registration number. The information in the registry entry suggests that your trial was registered after patient recruitment began. PLOS ONE strongly encourages authors to register all trials before recruiting the first participant in a study.

a) your reasons for your delay in registering this study (after enrolment of participants started);

b) confirmation that all related trials are registered by stating: “The authors confirm that all ongoing and related trials for this drug/intervention are registered”.

Please also ensure you report the date at which the ethics committee approved the study as well as the complete date range for patient recruitment and follow-up in the Methods section of your manuscript.

**Additional Editor Comments (if provided):**

Please address the concerns from the reviewers particular the sample size and adjustment for multiple testings. The conclusion and the results should parallel each other

**Comments to the Author**

1. Is the manuscript technically sound, and do the data support the conclusions?

Reviewer #1: Yes

Reviewer #2: Partly

2. Has the statistical analysis been performed appropriately and rigorously? 

Reviewer #1: Yes

Reviewer #2: No

3. Have the authors made all data underlying the findings in their manuscript fully available?

Reviewer #1: Yes

Reviewer #2: Yes

4. Is the manuscript presented in an intelligible fashion and written in standard English?

Reviewer #1: Yes

Reviewer #2: No

5. Review Comments to the Author

Reviewer #1: This is a well-written description of an gait therapy intervention clinical trial. The primary outcome in these studies is always a bit squishy, but the investigators used appropriate nonparametric techniques. I have some concern about the sample size. Randomization is well-stated and the procedures for reducing bias seem appropriate.

1. The primary outcome analysis seems to be based on Friedman's test or ordinal regression (undefined: please include the specific model you are using), but this is not stated. It is not stated why "1" is a clinically relevant difference. It is not stated why "1" is a reasonable S.D. It is not stated what is being tested and how (if it's Friedman's test you cannot use the formula for a t-test). With such a small sample size, it is unlikely that the central limit theorem will take care of the normality, so this needs to be stated much more clearly.

2. In the conclusion section it should be stated whether the assumptions of the same size computation (e.g., S.D., effect size) were realized.

3. Your tables have a bunch a stars indicated significance at 0.05 but you have not made any attempt at adjustment for multiple testing. Unless these are purely exploratory hypotheses, you have not preserved your type I error rate.

Reviewer #2: This RCT aimed to determine whether 4 weeks of 60-90 min 4 days a week hybrid assistive limb device (HAL) compared to usual care improved walking function in stroke survivors with severe limitations to walking. At 4 weeks and 6 months there were no differences between groups. In the HAL group the HAL training resulted in longer distances walked than during conventional therapy.

Abstract

1. The abstract conclusion should relate to the main aim of the study. Eg there were no differences between groups – especially considering the extra time and resources the HAL training required.

Introduction

2. The introduction contains information that would be better described in the methods. Eg lines 96-105 and Figure 1.

3. More description of the prior clinical trial could be provided in the introduction in relations to the findings and need for this RCT.

4. Short and long term should be defined

Methods

5. There was a large number of participants that were ineligible. Could reasons for exclsuion be provided?

6. What was the body size limit for the equipment (line 156)

7. The randomization procedure is not sufficient described

8. It is not necessary to statistically test the difference in the table of characteristics (Table 2). The sample size calculations are not designed to test this.

Results

9. The text is quite repetitive of what is in the table and therefore could be shortened.

Discussion

10. The discussion is very long and would benefit from being shortened with greater focus on the main results of this study eg paragraph 2 should discuss the main results and why there might not have been differences between groups. The HAL group received more therapy time and sessions (as they also received conventional training) and HAL training required mostly 2 therapists. Despite this there were no significant differences between groups.

11. In the HAL group HAL training resulted in greater distance than in conventional training. Interestingly the HAL group walked on average 50% less during conventional training than the conventional group. Perhaps mention should also be made of this?

Conclusion

12. The conclusion should better reflect the results - that there were no difference between groups for any outcomes despite the extra sessions, time and resources required for the HAL training. An equivalence trial was not carried out – therefore the statement regarding equal improvement should be removed.

6. PLOS authors have the option to publish the peer review history of their article (what does this mean?). If published, this will include your full peer review and any attached files.

Reviewer #1: No

Reviewer #2: No

---

## [Author Response · Author response to Decision Letter 0]

30 Oct 2019

We are grateful for the constructive comments and questions from the editor and reviewers and have considered all points in the revised version. Please find attached our point-to-point response and new manuscript with and without track changes.

---

## [Editor Report · Decision Letter 1]

31 Oct 2019

PONE-D-19-15743R1

A randomized controlled trial incorporating an electromechanical gait machine, the Hybrid Assistive Limb, in gait training of patients with severe limitations in walking in the sub-acute phase after stroke

PLOS ONE

Dear Ms. Wall,

Thank you for submitting your manuscript to PLOS ONE. After careful consideration, we feel that it has merit but does not fully meet PLOS ONE’s publication criteria as it currently stands. Therefore, we invite you to submit a revised version of the manuscript that addresses the points raised during the review process.

ACADEMIC EDITOR: please put your responses in bold. the version provided here is not acceptable. also remove the phrase 'trial' and use the term study.

We would appreciate receiving your revised manuscript by Dec 15 2019 11:59PM. To enhance the reproducibility of your results, we recommend that if applicable you deposit your laboratory protocols in protocols.io, where a protocol can be assigned its own identifier (DOI) such that it can be cited independently in the future. For instructions see: http://journals.plos.org/plosone/s/submission-guidelines#loc-laboratory-protocols

We look forward to receiving your revised manuscript.

Kind regards,

Thanh G Phan, PhD

Academic Editor

PLOS ONE

Additional Editor Comments (if provided):

It is not clear how the authors have addressed the reviewers' comments. usually this is done in bold to highlight the changes. the response to reviewers was very brief. I would also suggest that the authors removed the reference to trial given trial registration was performed after recruitment.

---

## [Author Response · Author response to Decision Letter 1]

4 Nov 2019

We are grateful for the constructive comments and questions and have considered all points in the revised version. Please find our point-to-point responses in the Response to Reviewers file and the attached Manuscript with and without track changes. We have now put the responses in bold and changed the wording “trial” to “study”.

Amended Competing Interests Statement: As pointed out under “Financial Disclosure”, the HAL suits were provided by Cyberdyne Inc. Japan and therefore Yoshiyuki Sankai and Hiroaki Kawamoto are listed as Investigators in the Study protocol (Supporting Information, S2). This does not alter our adherence to PLOS ONE policies on sharing data and materials. Cyberdyne Inc. were not involved in the study design, data collection, analysis and interpretation of data, in writing the manuscript, or in the decision to submit the manuscript for publication. 

All relevant data are within the manuscript and its Supporting Information files.

Kind regards, 

Anneli Wall (On behalf of all authors)

---

## [Editor Report · Decision Letter 2]

23 Jan 2020

PONE-D-19-15743R2

A randomized controlled study incorporating an electromechanical gait machine, the Hybrid Assistive Limb, in gait training of patients with severe limitations in walking in the sub-acute phase after stroke

PLOS ONE

Dear Ms. Wall,

Thank you for submitting your manuscript to PLOS ONE. After careful consideration, we feel that it has merit but does not fully meet PLOS ONE’s publication criteria as it currently stands. Therefore, we invite you to submit a revised version of the manuscript that addresses the points raised during the review process.

As new editor of this submission, I have carefully read the manuscript, the amendments, previous versions and reviews, so I will not need further external peer review for render a decision. The manuscript is methodically sound, but need addittional major changes before publication. If you agree to do so, please make the suggested changes and resubmit so I may make a new (and most likely last) assessment. The whole purpose for conducting any randomized study is comparing outcomes between two different groups, so any mention to within group comparisons should be removed in tables and writing (results, discussion...). Please, make the appropriate changes to focus on between group comparisons. The results and conclusions are well presented, but please do not offer statements about additional outcomes not originally categorized; For instance, your conclusion state that “No significant between-group differences were found regarding any primary or secondary outcomes, but HAL training enabled longer walking distances during training.” The second part of the sentence seems to look for something positive in the experimental results. But please, be aware that the authors predefined primary and secondary outcomes. So any statement should be based on these results, not on whether the participants walked longer distance or not. Walked distance was not a predefined outcome, but a secondary analysis of outcomes. In the same line, please avoid to present or disused other outcomes than the ones protocoled and measured. Please, remove table 1 and 4 and describe these classifications in the manuscript text.

We would appreciate receiving your revised manuscript by Mar 08 2020 11:59PM. To enhance the reproducibility of your results, we recommend that if applicable you deposit your laboratory protocols in protocols.io, where a protocol can be assigned its own identifier (DOI) such that it can be cited independently in the future. For instructions see: http://journals.plos.org/plosone/s/submission-guidelines#loc-laboratory-protocols

We look forward to receiving your revised manuscript.

Kind regards,

Jose María Blasco, Ph.D.

Academic Editor

PLOS ONE

---

## [Author Response · Author response to Decision Letter 2]

6 Feb 2020

We are grateful for the constructive comments from the editor and have considered all points in the revised version. Please find attached our responses and new manuscript with and without track changes.

---

## [Editor Report · Decision Letter 3]

13 Feb 2020

A randomized controlled study incorporating an electromechanical gait machine, the Hybrid Assistive Limb, in gait training of patients with severe limitations in walking in the subacute phase after stroke

PONE-D-19-15743R3

Dear Dr. Wall,

We are pleased to inform you that your manuscript has been judged scientifically suitable for publication and will be formally accepted for publication once it complies with all outstanding technical requirements.

With kind regards,

Jose María Blasco, Ph.D.

Academic Editor

PLOS ONE
---

## [Editor Report · Acceptance letter]

18 Feb 2020

PONE-D-19-15743R3 

A randomized controlled study incorporating an electromechanical gait machine, the Hybrid Assistive Limb, in gait training of patients with severe limitations in walking in the subacute phase after stroke 

Dear Dr. Wall:

I am pleased to inform you that your manuscript has been deemed suitable for publication in PLOS ONE. Congratulations! Your manuscript is now with our production department. 

With kind regards,

on behalf of

Dr. Jose María Blasco 

Academic Editor

PLOS ONE